



# fv3gfs-wrapper: a Python wrapper of the FV3GFS atmospheric model

Jeremy McGibbon[1], Noah D. Brenowitz[1], Mark Cheeseman[1], Spencer K. Clark[1,2], Johann Dahm[1],
Eddie Davis[1], Oliver D. Elbert[1,2], Rhea C. George[1], Lucas M. Harris[2], Brian Henn[1], Anna Kwa[1], W.
Andre Perkins[1], Oliver Watt-Meyer[1], Tobias Wicky[1], Christopher S. Bretherton[1,3], and Oliver Fuhrer[1]

[1]Vulcan Inc., Seattle, WA
[2]Geophysical Fluid Dynamics Laboratory, NOAA, Princeton, NJ
[3]Department of Atmospheric Sciences, University of Washington, Seattle, WA

**Correspondence:** Jeremy McGibbon (jmcgibbon@vulcan.com)

**Abstract.** Simulation software in geophysics is traditionally written in Fortran or C++ due to the stringent performance requirements these codes have to satisfy. As a result, these codes are often hard to understand, hard to modify and hard to interface with high-productivity languages used for exploratory work. `fv3gfs-wrapper` is an open-source Python-wrapped version of NOAA's FV3GFS global atmospheric model, which is coded in Fortran. The wrapper provides simple interfaces to progress

the Fortran main loop and get or set state from the Fortran model. These interfaces enable a wide range of use cases such as modifying the behavior of the model, introducing online analysis code, or saving model state and reading forcings directly to and from cloud storage. Model performance is identical to the fully-compiled Fortran model, unless routines to copy state in and out of the model are used. This copy overhead is well within an acceptable range of performance, and could be avoided with modifications to the Fortran source code. The wrapping approach is outlined and can be applied similarly in other Fortran

models to enable more productive scientific workflows.

## 1 Introduction

FV3GFS (Zhou et al., 2019) is a prototype of the operational Global Forecast System of the National Centers for Environmental Prediction. It is used within the U. S. National Oceanic and Atmospheric Administration (NOAA) Unified Forecast System (UFS, https://ufscommunity.org/) for operational numerical weather prediction. It uses the Geophysical Fluid Dynamics Labo-

ratory (GFDL) Finite-Volume Cubed-Sphere Dynamical Core (FV3). FV3 solves the non-hydrostatic equations of atmospheric motion discretized on a cubed sphere using a finite volume scheme on a terrain-following grid with D-grid wind staggering (Putman and Lin, 2007; Harris and Lin, 2013). The model is written in Fortran and parallelized using a hybrid OpenMP/MPI approach, which allows for performant execution through compilation.

However, development of an atmospheric model using a low-level, strongly typed programming language with a small user

base has trade-offs. Libraries for interacting with cloud storage, performing physical or statistical analysis, and using machine learning are not as readily available or widely used as they are in high-level languages such as Python. A Python interface to





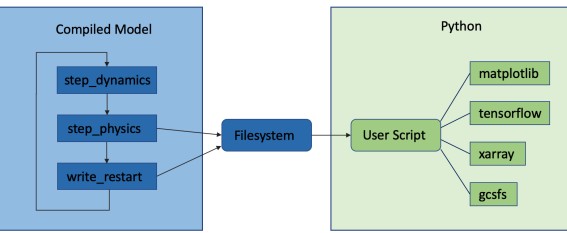

**Figure 1.** Schematic of Fortran-centric workflow using the filesystem to transfer data to Python user code.

the compiled Fortran code can enable a much larger Python user base to interact with this code, and allow a large ecosystem of Python tools to be interfaced with model routines.

Python is often integrated into Fortran modelling workflows as a post-processing tool, as shown in Figure 1. In this workflow,
Python is used to perform computations on data saved to the filesystem by the Fortran model. This approach has several shortcomings. It is rarely feasible to store the full-resolution model state at each model time step, so often statistics over time are stored instead. Unless sufficiently frequent snapshots are stored, computing new statistics directly from full-resolution instantaneous fields requires writing Fortran code to run in the model. This can be an issue if developer documentation is not available or the user is not familiar with Fortran. This approach requires writing to disk before data can be used in Python,
which may be unnecessary if the written data is not a necessary end product. Such filesystem operations can be a significant bottleneck in computation time. This approach also does not provide a way for Python to modify the behavior of the Fortran model. User code which modifies model behavior must be re-written in Fortran, for example when machine learning routines are trained in Python and exported to be used in a Fortran model (Ott et al., 2020; Curcic, 2019).

In this work, we present a Python wrapper for the FV3GFS global atmospheric model. As shown in Figure 2, the FV3GFS
model is compiled as a shared library with wrapper routines that provide an API to control and interact with the model. At the core of any weather or climate model is the main integration loop, which integrates the model state forward by a period if time. The wrapper splits the simple model main loop into a sequence of subroutines which can be called from Python. This allows the main loop to be written in Python, through calls to each section of the Fortran main loop (`step_dynamics`, `step_physics`). Furthermore, it allows copying variables into (`get_state`) or out of (`set_state`) the Fortran runtime
environment, so it can be used in Python functions which are also able to affect the integration of the Fortran model state.

As the wrapper currently stands, configuration is deferred entirely to the Fortran model code. The only change in initialization is that MPI is initialized by mpi4py, after which the MPI communicator is passed as a Fortran handle to the model initialization routines. This allows us to maintain feature completeness with the existing Fortran model, without re-writing configuration logic.

This Python-centric workflow enables a fundamentally different way to integrate tools available in the Python ecosystem into a Fortran modeling workflow. A user can add online diagnostic code after a physics or dynamics step, and perform input or output (I/O) to or from cloud resources. The model state can be reset to a previous one, allowing sensitivity studies to be



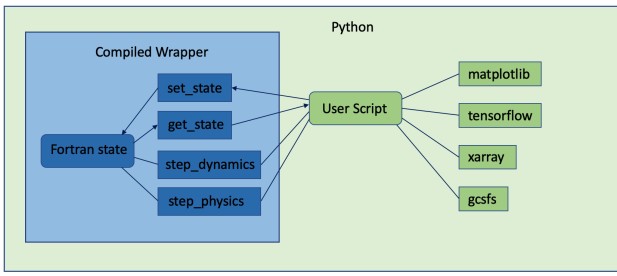

**Figure 2.** Schematic of Python-centric workflow using fv3gfs-wrapper, showing how it can interface with Python libraries during model execution.

run online. Custom logic can be added to the main loop after a physics or dynamics step, such as a machine learning corrector parameterization or nudging to a cloud-based forcing dataset. The use of a Python main loop makes it significantly easier to

integrate custom IO, diagnostic routines, and physical parameterizations into the model.

This ease of integration is an important tool when developing parameterizations using machine learning. When developing such schemes, offline performance does not guarantee performance when run online within a model. However it can be difficult to rapidly train a model in a language with machine learning libraries such as Python and then convert it for use in Fortran. Solutions have so far been based on Fortran executables, either by calling Python from Fortran (Brenowitz and Bretherton,

2019) or by re-implementing neural network codes in Fortran (Ott et al., 2020; Curcic, 2019). Because of the strong tooling available for machine learning in Python, it is an advantage to be able to include Python machine learning code within the atmospheric model. Presently Python code can only be integrated outside of the dynamics and physics routines, and not within the physics suite. Adding flexibility to introduce Python code between individual physics schemes remains a subject for future work.

Previous work exists using Python with compiled code to write atmospheric models and other high-performance parallel codes. qtcm (Lin, 2009) applies a similar wrapping approach to a quasi-equilibrium tropical circulation model using f2py (Peterson, 2009), an automated Fortran to Python interface generator. PyCLES (Pressel et al., 2015) is a full large-eddy simulation written in Cython, a variant of Python which compiles to C code and can interoperate with Python codes. CliMT (Monteiro et al., 2018) wraps Fortran model components into Python objects which can be composed to define a model main loop in

Python. In astronomy Python computational codes such as nbodykit (Hand and Feng, 2017) run using numpy (Harris et al., 2020) and MPI for Python (Dalcín et al., 2008), and are shown to scale to thousands of ranks. These previous works give confidence that a model using Python to call compiled code can provide the level of scaling and performance required for atmospheric and climate science research.

A consideration in designing new atmospheric models is the large amount of legacy Fortran code already available. As a

consequence, new model components are often written in Fortran so they can interface with such legacy code. Efforts to rewrite existing Fortran models, for example to run on GPU architectures, can benefit from the ability to progressively replace existing components with refactored or re-written codes in other languages.





To motivate the design choices made in this work, we present our main priorities:

- retain existing functionality of the Fortran model

- minimal sacrifice of performance

- easy to understand and modify the main time stepping loop

- minimal changes to Fortran code

Most of these priorities should be obvious, with a focus on improving model usability for researchers interested in modifying the behavior of the Fortran code. They would benefit from retaining the existing functionality they would like to modify, and

they should be able to easily understand how the code can be modified. They may require efficient model performance on high-performance computers for research problems using higher-resolution simulations. By minimizing the needed changes to the Fortran code, we can reduce the effort required to switch to a new Fortran model version.

While this wrapper has many applications, we will focus on illustrative scenarios relevant to our own FV3GFS model development work. In addition to reproducing the existing model behavior, we will show:

- augmenting the Fortran model with a machine learning parameterization

- include custom MPI communication as part of online diagnostic code

- perform online analysis in a Jupyter notebook

We will begin by showing in Section 2 how fv3gfs-wrapper can be used to reproduce bit-for-bit the results of the existing Fortran model. We will then show in Section 3 how fv3gfs-wrapper enables each of these use cases while achieving our

priorities of performance, ease of understanding, and ease of modification. Having presented the features of fv3gfs-wrapper by example, we will delve more deeply into their implementation in Section 4. Finally, we will discuss some of the challenges encountered in designing and implementing fv3gfs-wrapper in Section 4.5 before drawing our conclusions in Section 5.

## 2  Validation

For completeness and testing, fv3gfs-wrapper should be able to reproduce bit-for-bit the results of the Fortran model. This

allows us to test the logic wrapping the Fortran code. Because the wrapper executes identical Fortran code from the original Fortran model, bit-for-bit regression on one parameter configuration or forcing dataset gives us confidence the code can be used for any parameter configuration or forcing dataset. The implementation of this use case is as follows:

```
1  import fv3gfs.wrapper
2
3  if __name__ == "__main__":
4      fv3gfs.wrapper.initialize()
```





**Table 1.** Run times of examples and compiled Fortran model. Baseline refers to reproducing existing Fortran behavior. Examples were run for 6 hours of simulation time at C48 resolution on 6 processors on a 2019 Macbook Pro. Each example was run three times, and the shortest time is reported.

| Example | Run time (s) |
|---|---|
| Fortran baseline | 110 |
| Wrapper baseline | 110 |
| Random Forest | 116 |
| Minimum Surface Pressure | 110 |

```
5    for i in range(fv3gfs.wrapper.get_step_count()):
6        fv3gfs.wrapper.step_dynamics()
7        fv3gfs.wrapper.step_physics()
8        fv3gfs.wrapper.save_intermediate_restart_if_enabled()
9    fv3gfs.wrapper.cleanup()
```

The existing main routine in `coupler_main.f90` separates relatively cleanly into five routines: one each to initialize and finalize the model, one for dynamics (resolved fluid flow), one for physics (subgrid-scale processes), and one which will write intermediate restart data if intermediate restart files are enabled for the run and if we should write a restart on the current timestep. Each of these Python routines calls that section of the Fortran code, and then returns to a Python context.

The overhead of the Python time step loop and the wrapper functions is negligible in comparison to the computation done within a process (Table 1), meeting our performance goal. The conciseness of the main loop makes it easy to understand what the code is doing at a high level. This example is easy to modify, as shown in the use cases in the next section.

This code and the command-line examples below are available in the examples/gmd_timings directory of the git repository for fv3gfs-wrapper, using a 6-hour C48 run directory available at https://doi.org/10.5281/zenodo.4429297. The timings of each of these examples is included in Table 1. We can see the wrapper does not add significant overhead to the Fortran baseline timing.

## 3 Use cases in action

All examples discussed in this section are included in the public repository linked in the acknowledgements. We encourage the reader to download and run these examples on their own computer, using the example run directory available at https://doi.org/10.5281/zenodo.4429297.





## 3.1 Augmenting the model with machine learning

An important use case motivating this work is to be able to modify the operation of the model main loop, for example by adding a machine learning model which applies tendencies at the end of each timestep. This serves as an example for how the main loop

125 can be modified more generically, such as by adding I/O functionality or online diagnostics, using `fv3gfs.wrapper.get_state` and `fv3gfs.wrapper.set_state` to interface with the Fortran model.

```python
import fv3gfs.wrapper
import fv3gfs.wrapper.examples
import f90nml
from datetime import timedelta

if __name__ == "__main__":
    # load timestep from the namelist
    namelist = f90nml.read("input.nml")
    timestep = timedelta(seconds=namelist["coupler_nml"]["dt_atmos"])
    # initialize the machine learning model
    rf_model = fv3gfs.wrapper.examples.get_random_forest()
    fv3gfs.wrapper.initialize()
    for i in range(fv3gfs.wrapper.get_step_count()):
        fv3gfs.wrapper.step_dynamics()
        fv3gfs.wrapper.step_physics()

        # apply an update from the machine learning model
        state = fv3gfs.wrapper.get_state(rf_model.inputs)
        rf_model.update(state, timestep=timestep)
        fv3gfs.wrapper.set_state(state)

        fv3gfs.wrapper.save_intermediate_restart_if_enabled()
    fv3gfs.wrapper.cleanup()
```

The example includes a compact random forest we have trained on nudging tendencies towards reanalysis data. The separation of physics and dynamics steps in the code makes it clear that the machine learning update is applied at the end of a physics step, and is included in any intermediate restart data. The random forest model used in this example is trained according to the approach in Watt-Meyer et al. (2021), with a small number of trees and layers chosen to decrease model size. As a proof of concept, the example model has not been tuned for stability, and may crash if run for longer than 6 hours or using a run

directory other than the example provided. Model stability can be increased by enforcing the model specific humidity to be non-negative after applying the random forest update.

This example showcases how the wrapper makes it easy to modify the operation of the Fortran model. In our own efforts to re-write the FV3 dynamical core in a Python-based domain-specific language (DSL), we have directly replaced a call to the Fortran dynamics step with Python-based code. We have also added nudging routines which directly access Zarr (Miles et al.,

2020) reference datasets stored in the cloud, and IO routines to save model snapshots to Zarr files in cloud storage as the model





executes. With Python's threading support, this data transfer can happen as the Fortran code is running. These tasks would be difficult to implement in Fortran, due to more complex threading interfaces, no existing bindings for Zarr, and a lack of support from cloud storage providers.

## 3.2 MPI communication

When writing parallel models, inter-process communication is an important functionality. MPI4py (Dalcín et al., 2008) provides Python bindings for MPI routines, and supports the use of numpy arrays. Using MPI4py, we have been able to implement halo updates, gather, and scatter operations. The syntax for MPI4py is similar to the syntax used in Fortran. In our implementation, the same MPI communicator is used by the Fortran code as is used by MPI4py.

Here we show an example of computing the minimum global surface temperature and print it from the root process. This
is a relatively simple example showcasing how you can use MPI4py within the model to compute diagnostics using inter-rank communication.

```python
import fv3gfs.wrapper
import numpy as np
from mpi4py import MPI

ROOT = 0

if __name__ == "__main__":
    fv3gfs.wrapper.initialize()
    # MPI4py requires a receive "buffer" array to store incoming data
    min_surface_temperature = np.array(0)
    for i in range(fv3gfs.wrapper.get_step_count()):
        fv3gfs.wrapper.step_dynamics()
        fv3gfs.wrapper.step_physics()

        # Retrieve model minimum surface temperature
        state = fv3gfs.wrapper.get_state(["surface_temperature"])
        MPI.COMM_WORLD.Reduce(
            state["surface_temperature"].view[:].min(),
            min_surface_temperature,
            root=ROOT,
            op=MPI.MIN,
        )
        if MPI.COMM_WORLD.Get_rank() == ROOT:
            units = state["surface_temperature"].units
            print(f"Minimum surface temperature: {min_surface_temperature} {units}")

        fv3gfs.wrapper.save_intermediate_restart_if_enabled()
    fv3gfs.wrapper.cleanup()
```



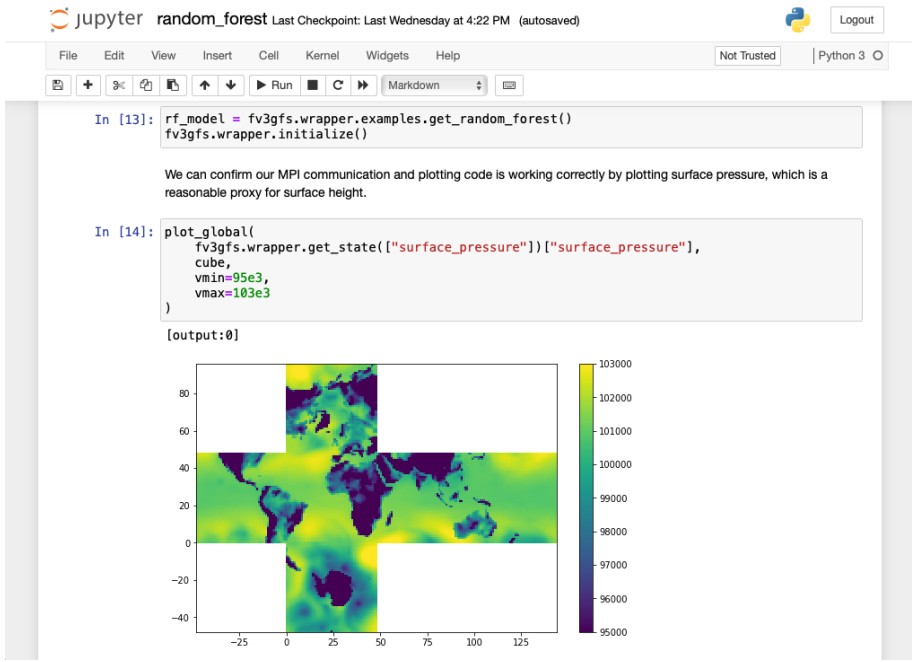

**Figure 3.** Screenshot of Jupyter notebook example using MPI communication to gather a field on 6 processes and plot it on the first process.

## 3.3 Interactive use in a Jupyter notebook

While we typically run the model using batch submission or from the command-line, all of the examples above can be executed from within a Jupyter notebook using ipyparallel. This allows retrieving, computing, and plotting variables from the Fortran model while it is paused at a point of interest. It also can serve as explicit documentation of modelled phenomena, whether to communicate to other model developers or for use in an educational setting.

We have prepared an example which inspects the machine learning example model, using MPI communication from Python to gather and plot variables on a single rank (Figure 3). It can be accessed in the `examples/jupyter` directory of the Github repository, and makes use of Docker to ensure portability. While the example is written to run on 6 processes, ipyparallel allows notebooks to be run at larger scales on high-performance computing (HPC) clusters if the configuration is modified appropriately.

## 4 Implementation

### 4.1 Information transfer

To augment the Fortran model, we need to be able to read from and write to its state. This information transfer can be done in two ways, either by providing an interface to copy data between Fortran and Python arrays (effectively C arrays), or using





the same memory in both codes. Re-using memory requires that the Fortran code use a pointer to a C array, allowing the same
pointer to be used by the numpy array on the Python side. In the Fortran code for FV3GFS, arrays used for physical variables are
defined as non-target `allocatable` arrays, which precludes sharing them with Python. It would require significant changes
to the Fortran code to instead use pointers to C arrays, which conflicts with our priority of making minimal changes to the
Fortran code. Instead, the getters and setters perform a data copy between numpy-allocated C arrays and Fortran arrays within
the wrapper layer.

## 4.2 Metaprogramming to pass arrays

Copying data from Python into a Fortran array requires independent code for each Fortran variable to provide an explicit
mapping between a Python variable or string and its variable name in Fortran. In our approach, each variable has two Fortran
wrapper subroutines for getting and setting that variable, logic within a C wrapper layer for calling those Fortran wrapper
subroutines, and header declarations for those subroutines.

Writing each of these manually would take significant time and effort. Instead we use Jinja templates to generate these
wrappers using JSON files declaring necessary information such as the Fortran variable name, standard name, units, and
dimensionality. For example, the Fortran variable name "zorl" has standard name "surface_roughness", units "cm", and di-
mensionality `["y", "x"]`. This greatly reduces the number of lines required to write the code. For physics variables, the
template file and data file are 89 and 459 lines, respectively while the generated Fortran file is 1894 lines. Physics variables are
also responsible for most of the lines in the 1680-line generated Cython file. Adding a new physics variable requires adding an
entry to a JSON file with its standard name, Fortran name, Fortran container struct name, dimensions, and units. This JSON
file is also used to automatically enable unit tests for the getters and setters of each physics variable.

## 4.3 Portability and testing using Docker

One choice made in developing fv3gfs-wrapper was to use Docker containers for testing and our own research use of the
wrapped model. It requires significant effort to properly install the dependencies of FV3GFS, Flexible Modeling System (FMS),
and Earth System Modeling Framework (ESMF) on a host system, and use the correct compiler flags and library setup to com-
pile the Fortran model. However, once defined the Docker container can be used identically on cloud computing resources, con-
tinuous integration systems, and our host machines without the need for separate configuration of the compilation process for
each system. Furthermore, it documents the process required to build the environment for the model and fv3gfs-wrapper, which
should help in setting it up directly (i.e. without use of containers) on a machine. Finally, it facilitates distribution of the model
to others who may not have access to HPC resources and may want to reproduce our results on personal computers or cloud
resources. The docker image at time of publication can be retrieved as `gcr.io/vcm-ml/fv3gfs-wrapper:v0.6.0`, or
from McGibbon et al. (2021b).



## 4.4 Extending this approach to other models

While we have applied this wrapping approach to the FV3GFS model specifically, nothing about it is particular to this model. Our methodology should generalize to other atmospheric and climate models. This wrapper is an example of how one can wrap Fortran models in general to be accessible through Python. While using Cython and Fortran wrapper layers (as we have done here) involves writing more code than automated wrapping tools such as f90wrap (Kermode, 2020), it provides the flexibility necessary to wrap the existing Fortran code with minimal changes. We found much of the repetitive boilerplate needed for this wrapping could be handled through Jinja templating. With this approach, the wrapper provides the flexibility required to interface with the potentially very complex build systems of existing Fortran models, and requires only minimal modifications to the existing Fortran, such as making variables and routines public to be accessed from the wrapper layer.

The use of getters and setters introduces a copy overhead cost when modifying the base model behavior. On the other hand, it avoids refactoring necessary for a shared memory implementation, which would require modifying the Fortran code to use C-accessible arrays which can be shared with Python. Writing a Fortran wrapper layer for the getters and setters ensures that any variable modifiable in Fortran can also be modified in Python.

In wrapping the FV3GFS, we have split the simple model main loop into a sequence of subroutines, which are then wrapped to call from Python. This task is likely to be different in other Fortran models, particularly models with abstract main loops or complex coupling infrastructures. So long as Fortran subroutines can be defined to execute each part of the model main loop, these can be wrapped to call from Python for model integration.

## 4.5 Challenges and limitations

Python reads many files on initialization when it imports packages. This can cause significant slow-down on HPC systems using shared filesystems. Approaches using parallel filesystems, such as Sarus on HPC or Docker-based cloud solutions, can avoid this issue. When a shared filesystem must be used, solutions exist such as python-mpi-bcast by Yu Feng, or modifying the CPython binary as reported by Enkovaara et al. (2011).

The wrapper currently treats the dynamics and physics routines each as a single subroutine. This does not allow inserting Python code within the physics suite, between schemes. This limitation may be removed in the future by adding a wrapper for physics schemes in the Common Community Physics Package (CCPP, Heinzeller et al. (2020)). Through CCPP, it should be possible to separate the physics driver into multiple calls, allowing Python code to be called between any chosen physics schemes.

It is also important to remember when trying to modify the behavior of FV3GFS that with or without a wrapper, it is still fundamentally a complex parallel model. Parallel code is difficult to test and can result in race conditions. It may be necessary to understand the physical relationships between different model variables in the Fortran code, and when they are updated. For example, a change to model specific humidity in FV3GFS requires a corresponding change to layer pressure thickness in order to conserve mass. To account for this, we have included a routine `fv3gfs.set_state_mass_conserving` which modifies layer pressure thickness according to any changes in water tracer amounts.



## 5    Conclusions

We have presented `fv3gfs-wrapper`, a Python-wrapped version of the FV3GFS atmospheric model. The wrapper allows users to control and interact with an atmospheric model written in Fortran. The simple and intuitive interface allows for a
Python-centric workflow and can be used to enable a wide range of use cases, such as machine learning parameterization development, online analysis, and interactive model execution. We do not see a decrease in model performance relative to the fully-compiled model, unless routines to copy state in and out of the Fortran model are used. This copy overhead is well within an acceptable range of performance, and could be avoided with modifications to the Fortran source code.

We showed examples of how Python and Docker can be used to reproduce and modify the existing Fortran model, and how
the Fortran code can be called in an interactive Jupyter environment. In addition to accelerating research and development workflows, these examples show how a full-fledged weather and climate model can be made available for reproducible science and teaching.

The wrapping approach is outlined and can be applied similarly in other Fortran models. The Python-wrapped FV3GFS atmospheric model shows the way for a new generation of weather and climate models, where the top-level control flow of the
model is implemented in a high-level language such as Python while the performance critical sections are implemented in a low-level, performant language. This is a powerful approach which has already been used in popular Python packages such as Numpy and Tensorflow. We hope to see this approach extended to other models, enabling more widespread access to Python tools in developing traditional Fortran models, and reducing the barrier to access for researchers and students interested in introducing online analysis code into these models.

*Code and data availability.*   Code for this project is available in on Github at https://github.com/VulcanClimateModeling/fv3gfs-wrapper tag v0.6.0 (McGibbon et al., 2021a), https://github.com/VulcanClimateModeling/fv3gfs-fortran tag gmd_submission (Heinzeller et al., 2021), and https://github.com/VulcanClimateModeling/fv3gfs-util tag v0.6.0 (McGibbon et al., 2021c). It is also available as a Docker image at gcr.io/vcm-ml/fv3gfs-wrapper:v0.6.0 (McGibbon et al., 2021b). The model forcing directory used to time the examples is available as McGibbon et al. (2021d).

*Author contributions.*   Jeremy McGibbon contributed the initial version of the wrapper and has led its development. Significant code contributions have been made by Noah Brenowitz, Oliver Watt-Meyer, Spencer Clark, Mark Cheeseman, Brian Henn, Tobias Wicky, Oliver Fuhrer, and Anna Kwa. All authors were involved in design discussions and provided feedback on the code. Jeremy McGibbon prepared the manuscript with contributions from co-authors.

*Competing interests.*   The authors declare that they have no conflict of interest.



*Acknowledgements.* We thank Vulcan, Inc. for supporting this work. We acknowledge NOAA-EMC, NOAA-GFDL and the UFS Community for publicly hosting source code for the FV3GFS model (https://github.com/ufs-community/ufs-weather-model) and NOAA-EMC for providing the necessary forcing data to run FV3GFS. Computations supporting this work were also supported by a grant from the Swiss National Supercomputing Centre (CSCS) under project ID s1053.



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
