# Peer review of "fv3gfs-wrapper: a Python wrapper of the FV3GFS atmospheric model"

_Geoscientific Model Development, 2021_

## Referee Comment (RC2)

The article **fv3gfs-wrapper: a Python wrapper of the FV3GFS atmospheric model** presents a Python wrapper for an atmospheric model written in Fortran. The wrapper method is detailed, with reasons behind each choice presented. This article is useful as it presents a tool that can improve the accessibility and reproducibility of scientific studies that make use of the FV3GFS model, and also because its presentation can act as a guide to other researchers and developers who use other large models written in Fortran. I believe this article may be suitable for publication once improvements are made to the referencing, figures, and grammar.

**1 Title and abstract**

The title is clear and appropriate. Although it contains two acronyms, the type of object that the acronyms are referring to are clear in context.

In the abstract, acronyms are not defined. In addition, on lines 5 and 7 the way the word "state" is used on these lines is grammatically incorrect. I suggest adding an article or even expanding it to be clear exactly what the authors are referring to, so that a developer or user who is not as familiar with the code or is new to using models would know what they're talking about.

**2 Issues and questions in the main text**

1. Line 12: FV3GFS acronym needs to be defined at its first use.

2. Line 16: Missing reference for Fortran.

3. Line 16: Missing reference for OpenMP/MPI.

4. Line 16: MP/MPI acronyms need to be defined.

5. Line 20: Missing a comparison element, a possible element that the authors will likely want to replace is included in all caps as an example : "...available or widely used in LANGUAGES LIKE FORTRAN as the are in..."

6. Line 22: Remove "Python", to instead say: "...enable a much larger user base to interact..."

7. Lines 32, 108, and 275: "which" should be "that", since it is not preceded by a comma. Or you can add a comma before "which". Examples:

   - "I reviewed a journal article that contained a lot of words."
   - "I reviewed a journal article, which contained a lot of words."

8. Line 32: This sentence is awkward, recommend rewording. Potential example: "For example, user code that modifies standard model behavior by providing training through machine learning routines must first be run

in Python, then exported for use by the model code, and finally the model code must also be modified to accept the trained input." This example could likely be improved by the authors, who perhaps could use my interpretation of their intended meaning to further clarify their example.

9. Line 50: Use either "IO" or "I/O" consistently throughout paper and define upon first use.

10. Line 56: "...Python, it is advantageous to be able to.."

11. Line 60: Awkward sentence structure. Recommend something like, "This is not the first time Python and high-performance compiled programs have been combined."

12. Line 65: "In astronomy, Python..."

13. Line 66: recommend replacing "give" with "provide"

14. Line 71: "...Fortran models (an action that may be taken to, for example, allow the code to run on GPU architectures) may benefit..."

15. Line 71: Define GPU

16. Lists near Line 75 and 85: Recommend formatting list as part of a sentence (adding a comma at the end of each non-terminal item and a period at the end of the terminal item).

17. Line 84: "...model behavior, we will show how to:"

18. List near Line 85: Recommend rewording the beginning of each item to follow logically from the end of line 84.

19. Lines 88 and 94: "...to reproduce, bit-for-bit, the results..."

20. Line 95: "...wrapper executes Fortran code identical to the original model, bit-for-bit..."

21. Line 115: Recommend pointing the reader to the "code availability" section here, so that they don't think you've forgotten to link to the wrapper repository.

22. Lines 115-116: "The timings for each of these examples are included..."

23. Line 119: The public repository information is not included in the acknowledgements, update this to the correct section name.

24. Code Examples: PEP8 compliance should be improved for all code examples. One obvious violation is the import order: `https://www.python.org/dev/peps/pep-0008/#imports`

25. Line 194: Potential bug/typo in method name: `MPI.COMM_WORLD.Get_rank()` or `MPI.COMM_WORLD.get_rank()`?

26. Line 204: "Github" should be "GitHub".

27. Line 218: "getters" and "setters" need to be defined, so that a non-experienced reader can understand what they mean.

28. Lines 222-224: Sentence spanning these lines is hard to follow. On the first two readings I thought it was merely a run-on sentence, but now I think it is a list that is just missing some articles. Recommend making this clearer.

29. Line 225: Need a reference for Jinja templates.

30. Line 237: "However, once defined, the Docker container..."

31. Line 248: "...involves writing more code that would have been necessary if an automated wrapping tool such as f90wrap (Kermode, 2020) had been used, it provides..."

32. Lines 250-252: Some parts of this sentence could be clearer. Perhaps: "With this approach, Python wrappers can be produced for very complex build systems with only minimal modifications (such as ensuring the necessary variables and routines externally accessible) to the existing model code."

33. Line 253: Recommend replacing "On the other hand" with "However", since the bookending phrase "On one hand" is not used.

34. Line 257: Recommend clarifying text to something like "...split the FV3GFS model main loop into..."

35. Line 264: Provide a reference for python-mpi-bcast. This can be done by providing a webpage reference to the GitHub repository.

36. Line 271: "...FV3GFS that, with or without a wrapper, it is..."

37. Line 272: Recommend providing references supporting the statement "Parallel code is difficult to test and can result in race conditions."

38. Line 272-275: These sentences are hard to follow, please rewrite.

39. Line 282: "...unless routines to copy the model state in and out..."

**3 Figures**

Figures 1 and 2 would be improved by supplying:

1. Blocks indicating where user input and output lie in the workflow

2. Arrows on all lines, double headed if need be

---

## Author Response (AR1)

Reponse to Reviewer 1

*   Readers may find the terminology "FV3GFS" confusing, in particular when referred to as the "prototype of the operational GFS". The term "FV3GFS" presumably refers to the initial releases v0/v1 of the atmospheric component (called fv3atm by NOAA) of NOAA's modeling system, embedded in the NEMS infrastructure at that time (e.g., https://vlab.ncep.noaa.gov/documents/2381028/3659001/FV3GFS+Public+Release+V1/37d9d3e8-bd99-b728-d1d7-49c2b007be38). The entire model was accordingly called NEMSfv3gfs, which later was changed into UFS (Unified Forecast System). This is mentioned briefly in the acknowledgements. In particular there, but possibly also in the introduction, it would be great if that distinction could be made more clearly.

We have added more detail to the acknowledgements (third line, L312) and introduction (fourth line, L16).

*   Line 39: Should this read "... it allows copying variables into (set_state) or out of (get_state) the Fortran ..." (it is the opposite in the text)?

Yes, it should, thanks. Corrected.

*   Line 87: see lines 253-256 below.

See reply below.

*   Figure 3 and line 207: The manuscript somewhat assumes that readers are familiar with the underlying grid of the FV3 model, i.e. that a cubed sphere has six tiles and that the code is written such that at least one MPI task per tile is required. Maybe this could be mentioned in the text.

Great suggestion, added this detail to Figure 3's description where the meaning should be most apparent.

*   Lines 2018-2019: The data copies return the data on the native grid. Would it be possible, if so desired by the user or the specific type of application, to perform regridding operations on the fly during this step (note: this may not make sense when applying machine learning models, but for other applications such as plotting etc it would).

The simplest way of achieving this would be to apply regridding to the data once it has been copied into the Python arrays. Doing it on the fly requires doing the regridding in the Fortran code and would significantly complicate the code and adding an external dependency to a regridding library.

*   Lines 225-232: The variable "database" encoded in json files ressembles in its contents and requirements the Common Community Physics Package (CCPP) metadata. Both contains attributes such as standard names, units, dimenions, etc. Since CCPP was selected as the operational physics interface in all UFS applications in the following years (post-GFSv16), and since the authors explicitly mention taking advantage of the CCPP infrastructure in the future, would it make sense to consider interoperability here, i.e. either adopt the CCPP metadata table format or provide tools to convert between the formats?

In the long term, we are very interested in Python bindings for CCPP. It would make sense to interoperate with CCPP metadata more closely when doing so. In the scope of the presented project, there isn't a practical use of converting between these formats. Since both CCPP metadata and the code-generator JSON files have fields the other does not have, it would require changing both metadata formats to have mappings in both directions. The concerns they fill are subtly distinct - CCPP needs to differentiate between individual schemes, while the JSON data documents the full model code. The JSON data must expose exactly how to retrieve a variable to be able to generate code to do so, while CCPP can handle this automatically without the use of metadata.

*   Lines 234-243. Since the prototype releases FV3GFS v0/v1, the portability of the UFS has improved considerably. This also includes the prerequisite libraries mentioned in the text. To date, the UFS community released the Medium-Range Weather Application v1.0 and v1.1, as well as the Short-Range Weather Application v1.0. All of these run on a wide range of platforms, including generic Linux and macOS, with detailed instructions and software package

s (NCEPLIBS-external, NCEPLIBS). Also, as part of the releases, ready-to-use Amazon Machine Images (AMIs) for use on Amazon's AWS are provided, for the SRW App v1.0 a Docker container is also provided.

These are great developments! We've modified this section to be more agnostic about the challenges of installing the base model. We would be happy to reference any use of Docker by the UFS, but we were unable to find a Docker image for the SRW Application on its github repository, documentation, or release announcements. The Dockerfile used to produce that image would be very helpful for our future work.

* Line 245. How independent is the wrapper from the model grid (horizontal and vertical)? Are the interfaces generic and the developers need to add implementations for their model?

The wrapper itself (when using code generation) is fairly few lines of code. It involves about 700 lines of Fortran templates and manual wrapper code, a 100-line Python script to generate the final wrapper code, and about 600 lines of JSON variable metadata. This amount of code enables getting and setting about a hundred different variables, as well as executing all parts of the Fortran model. Most of this would need to be re-written when applying the same wrapping approach to another model. We tried to be consistent in referring to the "wrapping approach" being applied to another model, as opposed to the wrapper itself. One could say the interfaces are generic and that developers need to add implementations, but some details about the interface may also make sense to change (e.g. in our model main loop, we have dynamics and physics, but one may have other components or want a finer-grain control over physics stepping).

* Lines 253-256: One alternative option that this reviewer has played with briefly is to use shared memory for the in-situ model variables and to provide an external process with information such as "variable x can be found at this location in memory". With the appropriate metadata (type/kind/size/dimensions), an external process could read this data while the model is running. This would also be useful wrt. NVRAM. Have the authors considered this idea, and if yes, what were the stumbling blocks?

We have only explored within-process memory sharing, such as described with having Python initialize a numpy array and then tell the wrapped Fortran code to make use of that memory address. The main use cases we designed this wrapper for require fine-grained control of the operation of the model. The in-situ model is not a great fit for that.

As far as within-process memory sharing, the issue we ran into was it requires extensive changes to the Fortran code, which we wanted to avoid. We have not considered running Fortran and Python alongside one another because this would require large amounts of new control logic written in Fortran, and because we expect within-process communication to be faster.

* Section 4.5: One important aspect for model development is debugging and profiling, for which dedicated tools exist (gdb, valgrind, Intel Parallel Studio, ARM Forge, tau, Score-P/Scalasca, ...). Can these tools still be used if the top-level routine is Python like in the current implementation?

Yes, though it takes a little bit of work. If you run Python with debugging symbols available (e.g. by installing python-dbg), then you can use pdb up to the point where the wrapped code is called, and then attach to the process with gdb. Many of the tools you mention more recently have added the capability to debug Pyhton code.

* Lines 281-283: The authors have shown that there is no performance degradation in their tests. These use a C48 grid with 6 MPI tasks, presumable on a single node. Have tests been conducted for larger runs across multiple nodes, and do the authors expect that this statements holds true even when going to very high / extreme scaling applications across entire HPCs?

The main issue to deal with for extreme scaling is a known issue with running Python at scale; if you are using a shared filesystem then import calls can cause significant overhead (Frings et al. 2013, https://computing.llnl.gov/sites/default/files/spindle-paper.pdf). This can be solved by using a non-shared filesystem on the nodes, for example using /dev/shmem, Sarus or Singularity images, or through modified Python binaries which avoid doing these filesystem accesses on each MPI rank, or with tools like Spindle (https://computing.llnl.gov/projects/spindle). After solving for t

his issue, we expect this statement to hold true at extreme scale.

We have not run such tests for this particular code, but we were able to scale a pure Python implementation of cubed-sphere halo updates up to 384 nodes on Piz Daint with no scaling falloff, and other groups have run parallel Python codes at much larger scale.

\* Lines 296 and 333: This reference is incorrect.

The reference does appear incorrect, but this is the suggested reference of the cited zenodo resource. The reference was auto-generated from the Fortran git sources, and so includes usernames for users who do not have full names listed, and can contain multiple entries for people who contributed with multiple git credentials. We would appreciate guidance from GMD on what to do with this citation.

\* Acknowledgements: see above for a clearer distinction of UFS, FV3GFS, fv3atm.

We have added the specific detail of when the model code was forked from UFS, to make this clear. Precise details are included in the referenced code, as git history.

\* References: Please check all references thoroughly, several of them have "wrong" author names etc (e.g. lines 345 and 350, also 333 as mentioned above)

As above, would appreciate guidance from GMD on what to do with references that have "wrong" author names which do exist on the referenced DOI.

\* General comment on language: The document uses American English as far as the reviewer can see, but in at least one place (line 24, "modelling workflows") British English is used. Please check.

Corrected, thank you.

Response to Reviewer 2

Title and abstract

- Added definitions of NOAA and FV3GFS
- replaced "state" with "variable"

Issues and questions in the main text

1. Acronym defined.
2. Citation added.
3. Citations added.
4. Acronyms defined.
5. Added.
6. Updated.
7. Updated where noted, and elsewhere in the document.
8. The sentence and the preceding one have been re-worked.
9. Fixed.
10. Updated.
11. Updated.
12. Updated.

13. Updated.

14. Updated, with slightly briefer phrasing.

15. Defined.

16. Reformatted lists as part of a sentence.

17. Updated.

18. Updated.

19. Updated.

20. Updated.

21. Added reference to the Code Availability section.

22. Updated.

23. Updated.

24. Placing the fv3gfs.wrapper import at the top of the file is done intentionally, because depending on the system configuration, the import can cause a segmentation fault if done after importing numpy. This happens if there is a conflict between libraries used to compile numpy and the wrapper. We ran all code examples through flake8 and found no errors. Based on the reviewer's suggestion we moved the standard library import to come before external library imports in the machine learning example.

25. This is not a typo, mpi4py provides capitalized routines which take in numpy arrays (or anything providing a buffer interface) which are low-level and efficient, and non-capitalized routines which take in arbitrary python objects and use pickle under the hood. Several non-communication routines like Get_rank() are also capitalized.

26. We assume the line number here is mis-labelled since github does not appear on this line. Updated to "GitHub" spelling in code and data availability section.

27. Defined "getters and setters".

28. These two sentences have been reworked and are hopefully more clear.

29. Added hyperlink to Jinja documentation

30. This text was removed in response to another reviewer comment.

31. To avoid adding another tense to the sentence, we fixed the grammatical error by saying "than using automated wrapping tools".

32. Replaced with the suggested wording.

33. Updated.

34. Updated.

35. Added a webpage reference.

36. Updated.

37. Added a reference and re-worded statement.

38. Re-worded the explanation in concrete terms, which should be easier to follow.

39. Updated.

Figures

1. This is either too simple or too complex to include in the diagram, depending on how you interpret "user input". The only user code included in the diagram is the "User script" block, so user input and output should be only to and from that block. If you mean "files provided by the user", this also includes files read by the Fortran model, which we don't want to document here. Depending on the routines used, the file handle open might occur in the user script or in a library such as xarray or gcsfs. However the place where the purpose of those files and how they are operated on is defined would be the user script. To keep the figures simpler, we would like to exclude I/O details from the figure and leave it up to the reader's imagination how they might perform I/O in their user script.

2. We experimented with several alternatives to arrow-heading on these two figures before settling on the final design. The initial design had arrow heads on all lines, as suggested by the reviewer. We felt this made the meaning of the arrowheads unclear, and decided instead to use arrows to emphasize the Fortran model main loop and inputs or outputs of the Fortran model. We have added a description of the purpose of the arrowheads to both figure captions.

Response to Reviewer 3

Thank you for the comments! They're certainly helpful. We'll clarify below what points are outside the scope of these revisions, but we will take all of these into account for our development. In general we welcome contributions and collaboration on our codebase. If you know someone who would like to make extensive use of the wrapper, we encourage them to get in touch with us via email or on github.

1. This is outside of the scope of this manuscript, but some information for your reference follows. The initialization logic for the model is fairly complicated, and we've decided to completely leave it up to the existing Fortran code for this iteration. Several iterations of the namelist documentation are available online. We cannot attest to how well it matches the particular code we are using, but documentation of namelist options for the dynamical core are available at https://www.weather.gov/media/sti/nggps/20160127_namelist_guide.pdf. While we aren't using CCPP, the CCPP documented namelist options in part overlap with our fork, and are documented at https://dtcenter.ucar.edu/GMTB/v4.0/sci_doc/CCPPsuite_nml_desp.html

Anything managed by a namelist parameter should not require re-compilation.

When we do write initialization code, especially while writing port of this model in a domain-specific language (GT4py), we will try to make the options available as documented as possible in those projects. But for the wrapper itself, we will likely continue to rely on the existing code for initialization.

2. See response to (1).

3. Added a line when get_state is introduced to mention units information being included. Line 24 of the "MPI Communication" example also showcases this. We're interested in adding some units-aware computation ability to the Quantity object, but this is out of the scope of this manuscript.

4. This is outside the scope of this manuscript, but this is at the top of our list of planned features to add. Quantity is documented in the fv3gfs-util documentation, where it is defined: https://fv3gfs-util.readthedocs.io/en/latest/state.html#quantity

5. This is outside the scope of this manuscript, but thank you for noting it. This happens because of importing a markdown file (the readme) into a restructured text file (the sphinx doc page).

6. Reduced the horizontal size of the image a bit, which should have the same effect as increasing font size (the image gets scaled to a certain width). Replaced all blues with lighter hues.

7. This is outside the scope of this manuscript, but we have some information for your reference. We previously used Singularity for this purpose on other projects (running the Fortran model directly, and running a pure-Python port of the dynamical core), but aren't planning to maintain Singularity builds at the moment. It can take significant effort to do so because Singularity requires you use the same library versions in your image as are available on the HPC system, otherwise the performance is significantly degraded.

If you have a colleague who plans to make more extensive use of the wrapper on HPC and would like this to be tested, please have them reach out to us directly and we may be able to set something up with a bit of collaboration. We don't currently require this functionality or have any collaborators who do.

While you need Google credentials to run some of our examples in the repository, you should not need them to run the examples included in this manuscript - the run directory used is available as a Zenodo download. If you prepare the run directory appropriately (copying your runfile into the supplied Zenodo run directory) and then execute the runfile, you should not need credentials. See the file at examples/gmd_timings/Makefile for more details.

---

## Author Response (AR2)

We would like to thank the editors and all reviewers for their contribution to this review process. Responses follow.

Report 3:

1) updated according to (2)
2) updated
3) it is unclear what change the reviewer is asking for here (filesystem operations are already named and filesystem operations are a particular type of I/O), so we have not made a change
4) updated
5) replaced "section" with "statement", as the data availability statement is not a numbered section
6) updated

Report 1 / Topical Editor comments:

The Zenodo references have been updated as best we can with full author names. There are still some usernames remaining for which we either cannot determine the author name (e.g. "John" from Amazon Web Services), or for which the author listed on the DOI is a software tool (pyup.io bot).

The Copernicus bibtex style file (bst) does not support the formatting requested for Zenodo entries, so we have updated our bst to reflect the requested output. There does not seem to be a way to upload this changed file in the upload form. If we cannot upload this file and you require these changes, please let us know and we can send them separately.